# Quaternary Ammonium Groups Modified Magnetic Cyclodextrin Polymers for Highly Efficient Dye Removal and Sterilization in Water Purification

**DOI:** 10.3390/molecules28010167

**Published:** 2022-12-25

**Authors:** Bingjie Liu, Shuoxuan Wang, He Wang, Yong Wang, Yin Xiao, Yue Cheng

**Affiliations:** 1School of Chemical Engineering and Technology, Tianjin University, Tianjin 300350, China; 2School of Science, Tianjin University, Tianjin 300350, China

**Keywords:** β-CD polymer, adsorbent, dye pollution, sterilization

## Abstract

Water recovery is a significant proposition for human survival and sustainable development, and we never stop searching for more efficient, easy-operating, low-cost and environmentally friendly methods to decontaminate water bodies. Herein, we combined the advantages of β-cyclodextrin (β-CD), magnetite nanoparticles (MNs), and two kinds of quaternary ammonium salts to synthesize two porous quaternary ammonium groups capped magnetic β-CD polymers (QMCDP1 and QMCDP2) to remove organic pollutants and eradicate pathogenic microorganisms effectively through a single implementation. In this setting, β-CD polymer (CDP) was utilized as the porous substrate material, while MNs endowed the materials with excellent magnetism enhancing recyclability in practical application scenarios, and the grafting of quaternary ammonium groups was beneficial for the adsorption of anionic dyes and sterilization. Both QMCDPs outperformed uncapped MCDPs in their adsorption ability of anionic pollutants, using methyl blue (MB) and orange G (OG) as model dyes. Additionally, QMCDP2, which was modified with longer alkyl chains than QMCDP1, exhibits superior bactericidal efficacy with a 99.47% removal rate for *Staphylococcus aureus*. Accordingly, this study provides some insights into designing a well-performed and easily recyclable adsorbent for simultaneous sterilization and adsorption of organic contaminants in wastewater.

## 1. Introduction

In every moment of our daily life, a variety of contaminants originating from industrial discharges, biological waste, agricultural effluent, and household garbage, pollute the water [1,2,3]. However, since water is a nonrenewable resource on which mankind depends for survival, the pollutants we dispose of in water will unavoidably infiltrate the hydrologic cycle and jeopardize human health if adequate treatment for toxic compounds is not implemented [4]. Dyes, among all kinds of chemical pollutants, have contributed a greater deleterious impact on both aquatic ecosystems and human lives due to their inclusion of extremely recalcitrant and intricately structured molecules [5,6]. For one thing, dyes with a certain concentration in water bodies can consistently hinder sunlight from penetrating aquatic flora because of their hard-to-decompose property, which will cause erratic photosynthesis, leading to oxygen depletion and aquatic ecosystem abnormalities. For another thing, dye molecules tend to have poor biocompatibility and even be highly toxic to human bodies, such as aromatics, which are recognized to be mutagenic and carcinogenic [7,8,9]. Additionally, the presence of pathogenic microorganisms in wastewater might have a negative impact on human health and demands for extreme caution [10]. Therefore, it is necessary to create practical technologies to remove these contaminants and remediate the water resource.

Adsorption is perceived as the most effective strategy to decontaminate dyes [11]. However, conventional adsorbents, such as activated carbons, have some unavoidable setbacks, including the inability to effectively remove certain dyes, costly regeneration, and inappropriate disposal of adsorbent residue [12,13]. Moreover, the disinfection process of actual water, such as chlorination, will inevitably generate detrimental disinfection byproducts and cannot be combined with the commonly used adsorption process, requiring high construction investment and laborious procedures [14,15]. Therefore, three factors need to be satisfied in the developing of new absorbents: (1) Higher recyclability, which can reduce secondary pollution and cost; (2) Multifunction, such as simultaneously enabling and rapidly removing multiple target pollutants; (3) Excellent adsorption abilities.

To satisfy requirements (1) and (3), we initially introduced magnetite nanoparticles (MNs) and constructed porous magnetic cyclodextrin polymers (MCDPs) using the TFTPN cross-linking approach [16]. Then, the surface of the MCDPs was modified with two kinds of quaternary ammonium salts (one is epoxypropyl trimethyl ammonium chloride, EPTAC for short, and the other one is epoxypropyl hexadecyl dimethyl ammonium chloride, EPCDAC for short) to obtain quaternary ammonium groups modified magnetic cyclodextrin polymers (QMCDPs), respectively, which promised the material a second function and improved its absorbing performance. The synthesis process is shown in Figure 1. In this design, β-cyclodextrin-based polymers were chosen to be the main part of our material. β-cyclodextrin (CD) has been applied to environmental remediation because of its ability to formulate a stable inclusion with different organic pollutants adducting through host–guest interactions, as well as its non-toxic, biodegradable, cheap, and easy-to-obtain properties [17,18,19]. Cyclodextrin polymers (CDPs), formed by the cross-linking reaction between the CD and the cross-linking agent, are materials that focus on the improvement of specific surface area and adsorption properties [20,21]. Moreover, they can be easily enhanced with different functional groups and achieve enhanced adsorption performance, as well as other extension applications [22]. Quaternary ammonium salt has a good biological activity as an antibacterial material, which can effectively kill a variety of microorganisms without causing drug resistance [23].

As a result, firstly, the QMCDPs exhibited a high degree of integration and synergism of CD, magnetite nanoparticles, and porous polymers. They did not only maintain the macrocyclic structure and multiple functional groups inherent to CDs, but also possessed developed porous structure, insolubility, and magnetism, which is inherent to magnetic polymers, and helpful in improving the ability of removing pollutants and recycling water [24,25,26]. Secondly, the quaternary ammonium group was connected to the cross-linked CDP-endowed QMCDPs’ electro-positivity, making it conducive to enhance the adsorption capacity of MCDP for anionic dyes [27]. Moreover, quaternary ammonium salt with long hydrophobic alkyl chains (QMCDP2) can interact with hydrophilic groups of bacteria, which can change the permeability of a bacterial cell membrane and produce cytolysis, leading to bacterial apoptosis [28]. The performance of QMCDPs in removing dyes from water was verified by adsorption experiments of two anionic dyes Orange G (OG) and Methyl Blue (MB) (Appendix A). In addition, *Staphylococcus aureus* was used as the representative for antibacterial experiments to verify the ability of QMCDP to exterminate bacteria in water.

## 2. Results and Discussion

### 2.1. Characterization of Materials

In this part, we used a few analytical technologies to demonstrate that our materials have successfully synthesized as we have designed, including Fourier transform infrared spectrometry (FT-IR), elemental analysis (EA), thermogravimetric analysis (TGA), ζ-potential, N_2_ adsorption isotherm (BET) and so on.

#### 2.1.1. FT-IR

According to the infrared spectra presented in Figure 1, the peaks at 3438 cm^–1^, 2923 cm^–1^, 2155 cm^–1^ and 1035 cm^–1^ are assigned to the –OH stretching vibration, the aliphatic –CH_2_ tensile vibration, the C–O–H and C–O–C stretching vibration of the glucose units in β-CD, while the resonations at 2243 cm^–1^, 1635 cm^–1^ and 1469 cm^–1^ are C≡N, C–C and C–F stretching vibrations of TFTPN, indicating that β-CDs were successfully cross-linked by TFTPN. The peaks at 1678 cm^–1^ in the infrared spectra of QMCDP1 and QMCDP2 are derived from –C–N stretching vibration, indicating that quaternary ammonium salts were successfully introduced into MCDPs. Specifically, it can be observed that the peak intensity of the aliphatic −CH_2_ tensile vibration at 2923 cm^–1^ in the infrared spectrum of QMCDP2 is significantly enhanced compared to that of MCDP and QMCDP1, demonstrating that the quaternary ammonium salts with long hexadecyl chains (EPCDAC) were successfully introduced into MCDP.

#### 2.1.2. EA

The results are shown in Table 1. The quaternary ammonium group content of QMCDP1 and QMCDP2 were calculated using Formula (1):(1)$$n_{QAS}=\frac{w_{N}\left(\mathrm{QMCDP}\right)-w_{N}\left(\mathrm{MCDP}\right)}{M_{N}\cdot 10^{-1}}$$
where *n_QAS_* (mmol·g^−1^) represents the quaternary ammonium group content, *w_N_*(QMCDP) (%) and *w_N_*(MCDP) (%) is the content of nitrogen in QMCDP and MCDP, respectively, *M_N_* (g·mol^−1^) is the relative atomic mass of nitrogen. The calculation results for QMCDP1 and QMCDP2 was 0.543 mmol·g^−1^ and 0.150 mmol·g^−1^, respectively. QMCDP1 obtains a higher content of quaternary ammonium groups, which might contribute to EPTAC’s smaller molecular structure, which can enter the interior of MCDP more easily and be combined with the bare hydroxyl groups in the narrow pores, while EPCDAC possessing a hexadecyl chain and can only be combined with hydroxyl groups on the surface or in larger pores of MCDP. Moreover, the contents of carbon and hydrogen in QMCDP2 are higher than that in QMCDP1 can also demonstrate EPCDAC’s decoration on QMCDP2.

#### 2.1.3. TGA and Zeta Potential

The thermogravimetric analysis and zeta potential test were used to demonstrate the thermal stability and surface electric charge of the materials, while providing side evidence that quaternary ammonium groups were successfully modified onto MCDPs.

Figure 2a shows the TGA curves of MN, MCDP, QMCDP1 and QMCDP2. The first weight loss of the four materials was at a low temperature (from 35 °C to around 130 °C), corresponding to the residual water [29]. When the temperature reached 270 °C, the covalent bond formed by the cross linking of TFTPN, which was broken and free functional groups were degraded, resulting in a mass loss of about 25% between 270 °C and 360 °C for MCDP, QMCDP1 and QMCDP2 [20]. The above results indicate that all the materials can maintain their own structural stability without obvious mass loss up to 270 °C, thus having sufficient thermal stability to accommodate the application environment for water purification.

When the temperature reached above 360 °C, the slow mass loss of the three MCDPs is mainly attributed to the alkyl chains in cyclodextrin monomers decomposing [30]. Additionally, the faster mass loss of both QMCDPs than MCDP indicated that the surface-modified quaternary ammonium groups of QMCDPs decomposed. The fourth weight loss observed at 690 °C could be attributed to the phase transition from the modified Fe_3_O_4_ to FeO and the deoxidation of FeO [31].

Moreover, the mass loss differences between 360 °C and 690 °C among the three materials could be used to compare the content of cyclodextrin, we can approximate the cyclodextrin content by calculating and comparing their differences, and the results are 17.348%, 17.472% and 25.447% for MCDP, QMCDP1, QMCDP2, respectively. Additionally, the higher weight loss of QMCDP2 may be attributed to the decomposition of the alkyl chains of EPCDAC.

MCDP and QMCDPs exhibited completely opposite zeta potentials, demonstrating that the positively charged quaternary ammonium groups were successfully modified to MCDP. Moreover, the negative potential of MCDP contributes to the active groups of the cyclodextrin and the cross-linking agent. The slight decrease in zeta potentials from QMCDP1 (38.2 mV) to QMCDP2 (32.0 mV) is due to the lower content of quaternary ammonium groups [32].

#### 2.1.4. Magnetism and the Content of MNs in MCDPs

The magnetism of QMCDP1 and QMCDP2 was investigated at room temperature by measuring the magnetization curve (Appendix A). The saturation magnetization of QMCDP1 and QMCDP2 was 13.76 emu·g^−1^ and 15.17 emu·g^−1^, and the coercive forces were 13.68 Oe and 11.70 Oe, respectively, showing superparamagnetic properties. Magnetic materials can be separated from the system in a simple and fast manner by magnetic separation after adsorption, thus improving the recoverability and reusability of the adsorbent.

Since the magnetism is relevant to the content of MNs, we used Atomic Absorption Spectroscopy (AAS) to detect the content of iron in MCDP, QMCDP1 and QMCDP2, respectively; the mass percentages of MNs in MCDP, QMCDP1 and QMCDP2 are 40.8%, 29.4% and 21.8% calculated following Equation (2) and the results can be found in Appendix A, which correspond to the magnetic properties of the material itself.
(2)$$w_{MN}\left(\mathrm{QMCDP}\right)=\frac{C_{Fe}\left(\mathrm{QMCDP}\right)\cdot M_{MN}}{3\cdot C_{QMCDP}\cdot M_{Fe}\cdot m_{QMCDP}}\times 100\%$$
where *w_MN_*(QMCDP) (%) represents the mass percentages of MN in QMCDP; *C_Fe_*(QMCDP) (mg·mL^−1^) represents the test result; *C_QMCDP_* is the concentration of test solution (mg·mL^−1^); *M_Fe_* (g·mol^−1^) and *M_MN_* (g·mol^−1^) are the relative atomic mass and molecular mass of iron, and MNs, *m_QMCDP_* (g) is the mass of QMCDP in the test solution.

#### 2.1.5. External and Internal Morphology

The external morphology of different materials could indicate that the MN particles have been successfully attached to the surface of MCDP, QMCDP1, and QMCDP2. The SEM image of MNs (Appendix A) showed that MNs had a spherical particle structure with diameters of 200 to 600 nm and typically rough surfaces, which could correspond to the particles on the “substrate” material in Appendix A, providing visual evidence of the existent form of MNs in the MCDPs.

For absorbents, the most intuitive test data to demonstrate their potential adsorption capacity are the nitrogen adsorption–desorption isotherm and pore size distribution (Appendix A). The BET (Brunauer–Emmett–Tell) specific surface areas (S_BET_) calculated according to the isotherms of MCDP, QMCDP1, and QMCDP2 are 107.482 m^2^·g^−1^, 31.759 m^2^·g^−1^, and 9676 m^2^·g^−1^, respectively. Additionally, the average pore diameters of MCDP, QMCDP1, and QMCDP2 are 4679 nm, 5736 nm, and 9146 nm, respectively. The increase in the average pore size of the three MCDPs and the decrease in the specific surface area contribute to the modification of quaternary ammonium groups on the surface of MCDP, which occupied the smaller pores.

On the other hand, Figure 3a,d indicate that MCDP contained many mesopores (2–50 nm), approximately a few dozen nanometers in size, and ultra-micropores. However, the transformation from MCDP to QMCDPs obviously reduced the number of mesopores on the surface, which could be seen as evidence that the quaternary ammonium salts have been successfully modified into MCDP and filled the pore structures of MCDP. Nonetheless, the surfaces of QMCDP1 and QMCDP2 are rough, and thus, could still exhibit a potentially porous and highly specific surface area, which is beneficial in increasing the contact opportunity of the adsorbent and dye, according to the following test results.

### 2.2. Adsorption Isotherm

#### 2.2.1. Anionic Dyes

This section investigates the effect of initial dye concentration on dye adsorption. To obtain the maximum adsorption capacity of the adsorbents and to explore the interaction between dyes and adsorbents, the equilibrium adsorption capacity of the adsorbent for OG and MB with initial concentrations of 0.1, 0.2, 0.3, 0.4, and 0.5 mM was studied. The maximal OG adsorption capacities of MCDP, QMCDP1 and QMCDP2 were determined to be 0 mg·g^−1^, 85.769 mg·g^−1^ and 34.775 mg·g^−1^, respectively. Additionally, the maximum adsorption capacities of MCDP, QMCDP1, and QMCDP2 for MB were 125.88 mg·g^−1^, 278.57 mg·g^−1^, and 174.50 mg·g^−1^, respectively. Compared to other adsorbents, the maximum adsorption capacities for the adsorption of MB to QMCDPs is relatively excellent considering the small specific surface area of QMCDPs (Appendix A).

The reason that MCDP has a larger surface area but a poorer adsorption performance needs to be clearly discussed. Additionally, it might correspond to the adsorption mechanism based on electrostatic interactions. The introduction of quaternary ammonium groups rendered the charges of the QMCDPs positive, so that they enhanced the adsorption capacity of two anionic dyes, OG and MB through electrostatic interaction [21].

Moreover, the reason that MCDP had a certain adsorption effect on MB but not on OG might be attributed to the different structure of the molecules leading to different electrostatic interactions with the absorbent (Appendix A) [33]. OG only possesses two anionic sulfonate groups, and both are on the same benzene ring, whose mutual repulsion will minimize the electrostatic contact between the dye and the adsorbent. In contrast, MB’s molecular structure has three sulfonate groups that are dispersed across different benzene rings; thus, the electrostatic connection between the dye and the adsorbent will be increased by the high anionic charge density, leading to a much greater adsorption capacity. We can conclude that the charge effect further enhanced the adsorption capacity of QMCDPs.

Moreover, the fact that the maximum adsorption capacity of QMCDP2 was smaller than QMCDP1 may be attributed to the hexadecyl chain on the quaternary ammonium group blocking the cavities of cyclodextrin, which elaborates for adsorption through host–guest interactions. Additionally, the hexadecyl chain might also weaken the interaction between the dye and the positive charge.

The equilibrium adsorption capacity of QMCDPs on OG approached its maximum adsorption capacity at a low initial concentration, and its equilibrium adsorption capacity was little affected by the change of the initial concentration (Figure 4a). By contrast, the equilibrium adsorption capacity of the adsorbent to MB obviously increased with the increasing initial concentration, and then slowed down (Figure 4b). To better investigate the experimental results, the isotherm data were tested to fit with the Freundlich and Langmuir models (Table 2). It is obvious that the Langmuir model is more suitable for fitting the adsorption isotherm data with a good linear relationship by comparing the *R^2^*, indicating that the adsorption process was monolayered.

#### 2.2.2. Electrically Neutral Pollutant

In order to explore the retention degree of the electrically neutral pollutant adsorption capacity of QMCDPs, Bisphenol A (BPA) was selected as a representative and the maximum adsorption capacity of BPA by the three absorbents was measured. Figure 5 and Table 3 present the experimental data and fitting data. Likewise, the Langmuir model is more suitable for fitting the isotherm data; thus, the adsorption process of BPA was also monolayer adsorption.

The maximum adsorption capacities of MCDP, QMCDP1 and QMCDP2 for BPA were 94.650 mg·g^−1^, 52.287 mg·g^−1^, and 109.65 mg·g^−1^, respectively. The adsorption capacity of QMCDP1 for BPA was only 55.24% of that of MCDP, due a hindered contact between BPA and the surface of the adsorbent, to a certain extent, due to the introduction of positive charges. However, for QMCDP2, the introduction of a hexadecyl chain enhances the hydrophobicity of its surface, thus obtaining a stronger BPA adsorption capacity than MCDP. Based on the experimental results in this section, it can be concluded that the adsorption mechanism of QMCDPs may include the hydrophobic interaction and host–guest interaction of cyclodextrins [34,35].

### 2.3. Adsorption Kinetics

The effect of contact time on the adsorption amount was investigated by measuring the adsorption of OG and MB solutions with a concentration of 0.1 mM using QCMDP1 and QMCDP2, respectively. Figure 6 shows the variation curves of the adsorption amount versus time. QMCDP1 could reach a larger adsorption within 30 s, and the rate constants of QMCDP1 and QMCDP2 were 3.9291 g·mg^−1^·min^−1^ and 0.57483 g·mg^−1^·min^−1^ for OG adsorption, and 3.2045 g·mg^−1^·min^−1^ and 1.3175 g·mg^−1^·min^−1^ for MB adsorption, respectively. The adsorption process of the dye by QMCDPs was able to approach the equilibrium adsorption amount in 10 min, indicating that QMCDPs can remove dyes from water bodies rapidly. Additionally, the fact that the adsorption rate of QMCDP1 was larger than that of QMCDP2 might contribute to the hindrance of the hexadecyl chain in QMCDP2, which reduced the contact rate of the dye with QMCDP2′s surface.

The pseudo-first-order model and pseudo-second-order model were used to investigate the adsorption kinetic data, and the fitted data are presented above in Table 4. The pseudo-second-order model is more suitable for fitting the adsorption process (R^2^ > 0.99), indicating that adsorption plays an important role in several rate-determining steps, such as external diffusion, internal particle diffusion, and adsorption.

### 2.4. Influencing Factors of Adsorption

The pH of the solution and the ionic strength are parameters that we investigated in this section as potential influences on the adsorption process. Ionic concentration can have an impact on the solubility of pollutants and further affect interactions between adsorbents and contaminants, while pH can change the surface charge property of an adsorbent and the molecules of pollutants already present in an aqueous solution [36,37].

The effect of the initial pH value on the adsorption of MB and OG for QMCDPs were investigated in the pH range of 2.0–12.0. As Figure 7 shows, the adsorption of MB and OG could both perform well in such a wide range, and the removal efficiency of MB at pH 12 remained superior, being 96.84% for QMCDP1 and 97.39% for QMCDP2. As for OG, the removal efficiency has slightly decreased from 10.0 to 12.0, while still maintaining a high absorbance rate. The following components might provide an explanation for the results: (i) Due to the incorporation of quaternary ammonium groups, QMCDPs have positively charged surfaces, and these positive charges could persist even at basic pH levels [38]. As anionic dyes with anionic sulfonate groups, MB and OG may readily adsorb to the surface of QMCDPs due to electrostatic attraction; (ii) MB and OG have π electrons because they possess multiple benzene rings in their structure and, therefore, π-π stacking interaction could be formed between the aromatic backbone of the dyes and the cyclodextrin polymer [39]. These findings showed that both QMCDPs were quite effective at removing anionic dyes from aqueous solutions across a wide pH range, which was a crucial benefit for the actual use of anionic dye adsorption by QMCDPs.

The effect of ionic strength on adsorption was also studied to evaluate the applicability of the material prepared, and NaCl was selected to investigate the effect of ionic strength. As shown in Figure 3c,d, NaCl with a concentration changing from 0 to 0.5 mol/L had no significant influence on the adsorption of MB and OG for both QMCDP1 and QMCDP2. On one hand, the presence of salt would reduce the solubility of substances, which in turn promoted the hydrophobic interaction between target molecules and adsorbents; on the other hand, the salt-out effect would destroy the hydrogen bonds, thus weakening the adsorption [40]. The negligible impact of NaCl on MB and OG adsorption indicating that the salt-out effect and hydrophobic interaction enhanced by salt were not critical, which was consistent with most reported CD polymers [17,19].

### 2.5. Adsorption Regeneration

Five cycles of adsorption–desorption experiments were performed using a 0.1 mol·L^−1^ sodium hydroxide solution with ethanol solution as eluent, and the removal rate data for each cycle are shown in Figure 8. The removal rates of OG and MB by QMCDP1 in the fifth cycle were 93.77% and 78.60% of that in the first cycle, respectively. Since the interaction force between OG and QMCDP1 is mainly through charge interaction, OG can be easily eluted from QMCDP1, and the removal rate only decreased by 6.2% after five cycles. For QMCDP2, the removal rates of OG and MB by in the fifth cycle were 75.83% and 25.34% of that in the first cycle, respectively. This indicates that the hexadecyl chain of QMCDP2 hinders the elution of the dye to some extent, especially for OG. Due to the smaller molecular weight of OG, it is easier to bind to the hexadecyl chain, resulting in a significant decrease in the adsorption capacity of QMCDP2 on OG after five cycles.

### 2.6. Antibacterial Performance

We selected *S. aureus* to evaluate the antibacterial performance of QMCDPs. As shown in Figure 9, QMCDP2 killed nearly all of *S. aureus*, reaching a 99.47% sterilization rate after 6 h of contact when the dosage was 1 g·L^−1^. However, the antibacterial ratio of QMCDP1 was subpar, almost identical to that of MCDP at around 35%, illustrating that the primary antibacterial ability of QMCDPs corresponds to the hexadecyl chain.

Combined with fluorescent images of live/dead staining (Figure 10) in which the untreated *S. aureus* exhibited green fluorescence with a little red fluorescence, still included many live bacteria after the treatment of MCDP and QMCDP1. However, the density of fluorescent bacteria in the treated samples was significantly lower than that of the untreated sample, indicating that the pores of the MCDPs could absorb a certain number of bacteria and partly achieve the purpose of sterilization. While after the treatment of QMCDP2, the *S. aureus* showed weak green fluorescence and strong red fluorescence, illustrating that most of the bacteria were dead. The results are consistent with the plate count method results showed in Figure 9, demonstrating that QMCDP2 can kill bacteria through electrostatic interactions and perturbation of the long alkyl chains.

The antibacterial performance of QMCDP2 was further verified after separation from bacterial solution in order to prove its ability of reuse in practical application and to achieve the purpose of cost saving. In this section, QMCDP2 was firstly magnetically separated from the bacterial solution, washed three times with highly pure water, dried, and applied to the next antibacterial experiment by the plate counting method. Meanwhile, the antibacterial ability of QMCDP2 after dye-absorbing was verified to prove that QMCDP2 could achieve both dye adsorption and antibacterial functions together in practical application. QMCDP2 was treated with OG and MB solution, respectively, and then magnetically separated. As shown in Figure 11, QMCDP2 can remove 93.62% of *S. aureus* after regeneration, and can remove 90.50% and 86.95% of *S. aureus* after the adsorption of OG and MB, respectively. The bacteria that failed to be removed during the regeneration process occupied part of the positive charge on the surface of QMCDP2, and the absorbed OG and MB would also occupy part of the positive charge, which led to a slight decrease in the antimicrobial ability of QMCDP2. The results showed that QMCDP2 had good regeneration performance and the ability to achieve bifunction of pollutant adsorption and antibacterial simultaneously.

## 3. Materials and Methods

### 3.1. Materials

Tetrafluorophenonitrile (TFTPN) was 98% pure and purchased from Dibo (Shanghai, China). Epichlorohydrin was purchased from Jiangtian (Tianjin, China). THF and DMF were absolutely anhydrous and obtained from Zesheng (Anqing, China). Ethylene glycol and 2,3-epoxypropyltrimethyl ammonium chloride (EPTAC, 95% pure) and were purchased from Aladdin (Shanghai, China). Diethyl ether, ethyl alcohol, methylene chloride and potassium carbonate were purchased from Yuanli (Tianjin, China). β-cyclodextrin (β-CD, 98% pure) and other reagents were purchased from Heowns (Tianjin, China). The water used in the experiments was ultrapure water (18 MΩ). All reagents were used without further purification.

### 3.2. Synthesis of Magnetite Nanoparticles (MNs)

The magnetite nanoparticles (MNs) are synthesized as previously described [41]. FeCl_3_·6H_2_O (1.08 g, 4.00 mmol), trisodium citrate dihydrate (0.20 g, 0.68 mmol) and sodium acetate trihydrate (1.20 g) were firstly dispersed in ethylene glycol (20 mL). After being stirred vigorously for 30 min, the mixture was then sealed in an autoclave and heated to 200 °C for 10 h. Black products were obtained by magnetic separation after cooling to room temperature, which were then washed with ethanol and ultrapure water for several times. MNs were collected for the following synthesis process after drying at 50 °C under vacuum for 12 h.

### 3.3. Synthesis of Magnetic Cyclodextrin Polymer (MCDP)

β-CD (3.28 g), TFTPN (1.60 g), potassium carbonate (5.12 g) and MNs (0.164 g) were placed in a dry 250 mL three-necked flask and anhydrous THF/DMF mixture (9:1 *v*/*v*, 160 mL) was added under the protection of argon gas. After stirring at 85 °C for 36 h, the mixture was magnetically separated and the brown solid was obtained, which was then washed several times with ultrapure water, ethanol, and dichloromethane, respectively and dried for later use.

### 3.4. Synthesis of QMCDP1

The dried MCDP particles (0.10 g) were dispersed in 1% NaOH solution (1.0 mL) by ultrasonication to which EPTAC (0.28 g) was then added and stirred at 65 °C for 6 h. The products were washed with water and ethanol three times and dried in vacuum at 50 °C for 12 h.

### 3.5. Synthesis of QMCDP2

The synthesis of EPHDMAC referred to a previous method [42]. Epichlorohydrin (23.13 g) was slowly added with a drip funnel to hexadecyl dimethylamine (26.95 g) while stirring. The mixture was then heated to 80 °C and stirred for 1 h. After vacuum distillation to remove the unreacted epichlorohydrin, the crude product was washed with diethyl ether three times, and vacuum dried for the following modification of MDCP. The synthesis process is the same as QMCDP1.

### 3.6. Analytical Methods

SEM images were recorded on an Apreo S LoVac electron microscope (FEI, Hillsboro, OR, USA) operating at 5.00 kV, after a thin platinum film was sprayed on the sample. FT-IR spectra were collected on FTS3000 Fourier spectrophotometer (BIO-RAD, Hercules, CA, USA) using KBr pellets. Magnetic characterization was carried out on LakeShore7404 (LakeShore, Columbus, OH, USA). The UNICUBE elemental analyzer (Elementar, Langenselbold, Hessen, Germany) was used to access the mass fractions of elemental C, H, and N. The N_2_ adsorption isotherm was achieved by Autosorb-iQ2 (Quantachrome, Boynton Beach, FL, USA). Thermogravimetric analysis (TGA) with a heating rate of 20 °C min^–1^ was carried out to examine the thermal stability of the samples (STA409PC, NETZSCH, Selb, Bavaria, Germany). The ζ-potential of the materials was measured by Nano ZS (Malvern, Worcestershire, UK). AAS was carried out on ICE3300 (Thermo Scientific, Waltham, MA, USA).

### 3.7. Adsorption Experiment

Stock solutions of OG, MB, and BPA at concentrations of 0.1, 0.2, 0.3, 0.4, and 0.5 mM were prepared at first.

For the adsorption thermodynamic experiments, 10 mg of adsorbent with 10 mL of the stock solution was added to a 20 mL test tube, shaken immediately, and placed in a shaker at a shaking rate of 200 rpm for 30 min at 25 °C. The supernatant was taken after the magnetic separation of the adsorbent. Then, the concentrations of OG, MB and BPA were determined by colorimetric experiments, the absorption rate was measured using a UV spectrophotometer at 475 nm, 590 nm, and 275 nm, respectively. All experiments were performed in triplicate.

The quantity of dye entrapped on the materials (*q_e_*) was determined by Equation (2), the Langmuir adsorption isotherm model (Equation (3)), and the Freundlich adsorption isotherm model (Equation (4)), which were all used to fit the experimental data
(3)$$q_{e}=V\left(C_{0}-C_{e}\right)m$$
(4)$$q_{e}=\frac{q_{max}k_{L}c_{e}}{1+k_{L}c_{e}}$$
(5)$$q_{e}=k_{F}c_{e}^{n}$$
where *V* is the volume of dye (L), *m* is the mass of material employed (g), *C*_0_ stands for the concentration in the liquid phase at the earliest stage (mg·L^−1^), and *C_e_* indicates the liquid phase dye concentration at equilibrium (mg·L^−1^).

For the adsorption kinetics experiments, 50 mg of adsorbent with 50 mL of stock solution at a concentration of 0.1 mM was added to a 100 mL conical flask, shaken immediately, and placed in a shaker at 25 °C with a shaking rate of 200 rpm. The 2 mL suspension was removed by syringe at 30 s, 1 min, 2 min, 5 min, 10 min, 20 min, and 30 min, respectively. The concentration of the dye in the supernatant was measured using a UV spectrophotometer after the magnetic separation of the adsorbent. The experimental data were fitted using the nonlinear form of the pseudo-first-order kinetic model (Equation (5)) and the pseudo-second-order kinetic model (Equation (6)):(6)$$q_{t}=q_{e}\left(1-e^{-k_{1}t}\right)$$
(7)$$q_{t}=\frac{q_{e}^{2}k_{2}t}{1+k_{2}q_{e}t}$$
where *q_t_* (mg·g^−1^) is the amount of organic pollutant adsorbed at the moment *t* (min); *k*_1_ (min^−1^) is the pseudo-first-order rate constant; *k*_2_ (mg·g^−1^·min^−1^) is the pseudo-second-order rate constant; and *q_e_* (mg·g^−1^) represents the amount of organic pollutant adsorbed at equilibrium.

The effect of solution pH and ionic strength on adsorption was investigated according to a previous paper [21]. In detail, 30 mg of QMCDP was added to 30 mL of pollutant solution (0.1 mmol L^−1^.), which was shaken for some time to reach equilibrium (according to the kinetic experiment). The solution pH was adjusted using 0.1 mol L^−1^ of HCl and NaOH aqueous solutions in the range of 2.0–12.0. The ionic strength was represented by NaCl aqueous solution ranging from 0.1 to 0.5 mol L^−1^. The removal efficiency of dye (*R*, %) was calculated using Equation (8)
(8)$$R_{dye}\left(\%\right)=\frac{C_{0}-C_{e}}{C_{0}}\times 100$$
where *C*_0_ (mg L^−1^) is the initial dye concentration in the solution; *C*_e_ (mg L^−1^) is the final or equilibrium concentration of dye in the solution.

In the regeneration experiments, after the magnetic separation of the adsorbent after the adsorption of the dye, the adsorbent was regenerated by desorption using aqueous NaOH solution with a concentration of 0.1 mol·L^−1^, water and ethanol in turn. The washed and regenerated adsorbent was dried and used for the next cycle of adsorption experiments. The cycle was repeated five times and the concentration of residual dye in each solution was measured using a UV spectrophotometer. Each set of experiments was repeated three times and the mean and standard deviation are shown in the data graph.

### 3.8. Antibacterial Experiment

#### 3.8.1. Plate Counting Experiment

All glassware used for the tests were sterilized in an autoclave at 121 °C for 60 min prior to use. All materials were sterilized by exposing to UV radiation for 60 min prior to the tests. MCDP, QMCDP1, and QMCDP2 (1.0 mg) were, respectively added to 1000 μL of the freshly prepared bacterial suspension. After mixing, bacterial suspensions were incubated at 37 °C in shaking incubator (Model G-25, New Brunswick Scientific Co., Inc., Edison, NJ, USA), at 200 rpm for 6 h. Then, 100 μL of the bacterial suspension was removed and serially diluted down to 10^−6^ of the original level. Individual 10 μL aliquots of each diluted bacterial suspension were then smeared onto the LB broth medium for total plate counting. After incubating at 37 °C for 18 h, the number of colonies were counted, and the results were expressed as mean colony forming units per volume (CFU/mL).

To verify the reusability of QMCDP2 after a single antibacterial experiment, QMCDP2 was magnetically separated from the bacterial suspension, and washed three times with ultrapure water. Moreover, the antibacterial ability of QMCDP2 after adsorbing dyes was further verified. QMCDP2 (10 mg) was firstly treated with 10 mL OG and MB solution (0.1 mM), respectively, then magnetically separated and washed three times with ultrapure water. Additionally, the antimicrobial experiments were repeated according to the previous steps.

#### 3.8.2. Live Dead Bacteria Staining Test

Fluorescence photography was performed using the confocal laser scanning microscope (CLSM). Upon incubation with MCDP, QMCDP1, and QMCDP2 with 6 h apart, the *S. aureus* was washed twice with a PBS solution and resuspended in 0.85% NaCl. It could be detected on CLSM after staining with LIVE/DEAD Bacterial Viability Kit (Thermo Fisher Scientific Inc.) for 15 min in the dark.

## 4. Conclusions

In summary, porous cyclodextrin polymers with magnetism were firstly obtained by adding magnetite nanoparticles during the cross-linking of cyclodextrins with TFTPN. The quaternary ammonium salts were then covalently modified on the surface and in the pore of the polymer, and obtained QMCDPs successfully. The following conclusions were drawn from our study:A series of characterizations demonstrated the successful modification of quaternary ammonium salts and the introduction of positive charges, which are beneficial to anionic dyes’ removal. The maximum adsorption of QMCDP1 and QMCDP2 can reach 85.769 mg·g^−1^ and 34.775 mg·g^−1^ for OG and 278.57 mg·g^−1^ and 174.50 mg·g^−1^ for MB, respectively, which are both significantly improved relative to the MCDP, which is without quaternary ammonium groups modification.Two quaternary ammonium salts with a different length of alkyl chains were introduced to the material, and the one with longer alkyl chain (QMCDP2) removed 99.47% *S. aureus* from the aqueous environment. Moreover, it still had good bactericidal ability after regeneration, achieving adsorption and antibacterial abilities at the same time, which indicates that QMCDP has great potential and broad application prospects as an antibacterial adsorbent in the field of water treatment.Multiple interactions are involved in the adsorption process. The anionic dyes’ removal might be mainly through electrostatic interactions, combined with host–guest interaction in which the quaternary ammonium groups, modified on the surface, took charge. While the adsorption of BPA is primarily through the hydrophobic effect combined with the host–guest interaction, the hydrophobic groups on the surface played the leading role.

Overall, the QMCDPs are promising in the application of purifying actual water bodies, which provides insights into designing effective and readily recyclable adsorbents for simultaneous sterilization and adsorption of anionic contaminants in wastewater.

## Data Availability

The research data in this study are available from the corresponding author upon reasonable request.

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
