# Peer review of "Quaternary Ammonium Groups Modified Magnetic Cyclodextrin Polymers for Highly Efficient Dye Removal and Sterilization in Water Purification"

_molecules, 2022, doi:10.3390/molecules28010167_

Round 1

Reviewer 1 Report

1. The innovation of this paper needs to be improved, and cyclodextrin as an adsorption material combined with magnetic nanoparticles or quaternary ammonium salt has been reported in the literature. The novelty and research or application value of this work should be highlighted.

2. In Figure 5, it is difficult to observe the difference between the various materials from the SEM Tmap of the different materials, and the image at a larger magnification should be given.

3. The discussion of the removal mechanism should be further advanced.

Author Response

Dear reviewer:

Thank you for your decision and constructive comments on this manuscript. We have carefully considered the suggestion and tried our best to improve and made some changes in the manuscript. And the point-by-point response of your comments is in the attachment.

We sincerely hope that you find our responses and modifications satisfactory and hope that the manuscript is now acceptable for publication.

Yours,

Bingjie Liu

Reviewer 2 Report

This manuscript reports the synthesis of a modified magnetic β-cyclodextrin polymer to remove organic dyes as well as disinfection from water. The literature review and starting hypothesis to develop synthesis route is appropriate. The data that was presented confirms the synthesis of the material and the characterization details are well described. However, there are some weaknesses in this work, such as incomplete experimental design, lack of depth in the results and discussion. The authors are encouraged to more carefully discuss their results in the context of the adsorption literature. A number of issues need to be addressed and are listed as follows:

1.Hydrophobic cavity of β-cyclodextrin plays an important role in the process of adsorbing pollutants, so what is the content of β-cyclodextrin in the material?

2. The content of magnetite of MCDP, QMCDP1 and QMCDP2 should be characterized.

3.The experimental design is not complete enough, and experiments on the influence of various hydrochemical conditions such as solution pH and ionic strength on dye adsorption should be supplemented

4.The authors are encouraged to review the recent literatures here more carefully about the adsorption properties of the material, such as adsorption capacity, adsorption rate, etc.

5. More detailed discussion on adsorption mechanisms should be explicated.

Author Response

(The authors gave the same response as above.)

Round 2

Reviewer 1 Report

The author has completed most of the modification work, and also needs to pay attention to the quality of images (such as SEM data map). Now I agree to recommend publishing this manuscript.

Reviewer 2 Report

I think the paper has been carefully revised and meets the publishing requirements